# LSPO: Length-aware Dynamic Sampling for Policy Optimization in LLM Reasoning

## Abstract

Since the release of Deepseek-R1, reinforcement learning with verifiable rewards (RLVR) has become a central approach for training large language models (LLMs) on reasoning tasks. Recent work has largely focused on modifying loss functions to make RLVR more efficient and effective. In this paper, motivated by studies of overthinking in LLMs, we propose Length-aware Sampling for Policy Optimization (LSPO), a novel meta-RLVR algorithm that dynamically selects training data at each step based on the average response length. We evaluate LSPO across multiple base models and datasets, demonstrating that it consistently improves learning effectiveness. In addition, we conduct a detailed ablation study to examine alternative ways of incorporating length signals into dynamic sampling, offering further insights and highlighting promising directions for future research.

## 1 Introduction

Since the release of ChatGPT, large language models (LLMs) have rapidly evolved beyond traditional natural language tasks such as summarization and question answering, extending into broader domains of reasoning and problem solving Chen et al. (2023); Yao et al. (2023); Chen et al. (2024a;b); Jaech et al. (2024); Huang & Yang (2025). A growing body of research has therefore focused on how to make LLMs more effective and efficient on reasoning-intensive tasks.

While a prominent line of work have investigated how to improve test-time performance without retraining a model from scratch Wang et al. (2022); Chen et al. (2023); Madaan et al. (2023); Gou et al. (2023); Guan et al. (2025); Chen et al. (2025), more recently, following the release of Deepseek-R1 DeepSeek-AI et al. (2025), reinforcement learning with verifiable rewards (RLVR) has emerged as a powerful and promising paradigm for post-training LLMs, yielding significant gains in reasoning ability. While Deepseek-R1 adopted GRPO Shao et al. (2024), subsequent research has proposed a variety of alternative algorithms to further enhance reasoning capabilities.

In general, efforts to improve RLVR training have explored several dimensions, including alternative loss functions, improved datasets, and better hyperparameter tuning. Most of the recent progress, however, has centered on loss design Zheng et al. (2025a); Wu et al. (2025), where specialized objectives are introduced to target phenomena such as long-context reasoning Zheng et al. (2025a) or length-awareness Wu et al. (2025). Although some works have also considered dynamic sampling strategies, these primarily aim at training efficiency rather than improving the final policy itself.

In this paper, we address this gap by introducing a novel approach to dynamic data sampling during RLVR, with the primary goal of improving model effectiveness, measured by the accuracy of the final trained model on test sets. Motivated by recent analyses of "overthinking" behaviors in reasoning models that highlight a strong correlation between response length and output quality, we propose **L**ength-aware **S**ampling for **P**olicy **O**ptimization (LSPO). LSPO is a meta-RL algorithm that adaptively filters training prompts based on the average response length of each question. As illustrated in Fig. 1, LSPO retains a fixed ratio of the shortest and longest responses, thereby focusing optimization on data most likely to yield meaningful improvements.

We evaluate LSPO on multiple base models, challenging datasets, and underlying RLVR algorithms. Our experiments show that LSPO consistently produces better-trained final models than baseline approaches. Furthermore, although LSPO requires additional sampling to match the batch sizes of standard RL, it remains effective under the same training time, highlighting its efficiency.

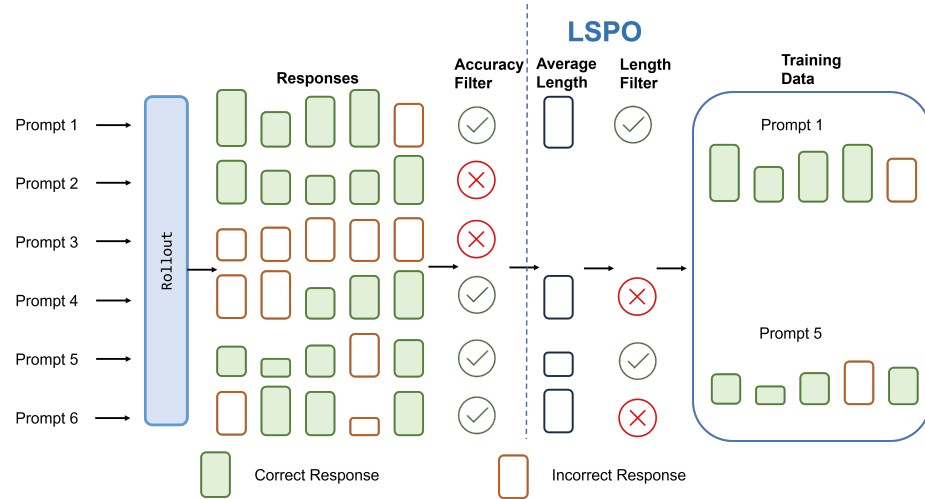

Figure 1: Illustration of Length-aware Sampling for Policy Optimization (LSPO). LSPO builds on common accuracy-based filtering and further filters responses by length, retaining prompts whose average response length is either the longest or the shortest. The vertical height of each response block represents its length.

As one of the first works to explore response length as a signal for data filtering in RLVR, we conduct an extensive ablation study to guide future research. Besides analyzing the hyperparameter choices of LSPO, we compare retention strategies such as value-based thresholds against fixed-percentile thresholds, and we also examine alternative filtering criteria such as accuracy.

Our main contributions are as follows:

- We propose Length-aware Sampling for Policy Optimization (LSPO), a novel meta–reinforcement learning algorithm that leverages response length as a signal for dynamic data filtering in RLVR.
- Through extensive experiments across multiple base models and RLVR algorithms on multiple datasets, we show that LSPO, as a dynamic sampling algorithm, can consistently improve final model effectiveness.
- We conduct a detailed ablation study on hyperparameters choices and filtering criterias to provide new insights into dynamic sampling for RLVR on reasoning tasks.

## 2 RELATED WORKS

**Reinforcement Learning with Verifiable Rewards for LLM Reasoning.** Since the release of GPT-o1 Huang et al. (2024), there has been growing interest in how to effectively train LLMs for reasoning tasks. The introduction of Deepseek-R1 DeepSeek-AI et al. (2025) demonstrated that reinforcement learning with verifiable rewards (RLVR) is a promising approach. Together with GRPO Shao et al. (2024), Deepseek-R1 has established a strong baseline that subsequent research has sought to improve. To address different limitations of GRPO, several alternative algorithms have been proposed. For instance, DAPO Yu et al. (2025) mitigates biased estimation and improves training efficiency, while GSPO Zheng et al. (2025a) targets challenges in long-context reasoning. A broader line of work has also focused on designing new loss functions tailored to specific objectives. Beyond loss design, researchers have also shown that other aspects of RLVR training can be optimized. For example, modifying sampling strategies to encourage better exploration Chen et al. (2024c); Zheng et al. (2025b) or adjusting the clipping ratio to stabilize learning and improve sample efficiency Yu et al. (2025). Our work focuses on dynamic sampling of training data to improve training effectiveness, offering a complementary direction to loss design.

**RLVR with Response Length** In relation to response length, prior studies have explored training models to generate shorter outputs Wang et al. (2025a); Yuan et al. (2025); Fatemi et al. (2025), pre-

dicting the length of responses Aggarwal & Welleck (2025), or adaptively allocating computational effort based on task difficulty Zhang et al. (2025); Lou et al. (2025). In contrast, our work leverages response length as a criterion for dynamic sampling in RLVR, an underexplored direction.

**Dynamic Sampling on RLVR Training** Dynamic sampling in RLVR has been studied primarily as a means to improve training efficiency. One line of work first trains on the full RL dataset and then identifies a smaller, more informative subset for continued training Li et al. (2025); Wang et al. (2025b). Another line filters data dynamically during RL training. At the rollout level, PODS Xu et al. (2025) downsamples trajectories that contribute the highest variance in reward for each prompt. At the sample level, DAPO Yu et al. (2025) applies dynamic sampling to discard policies with zero gradients. Later on, GRESO Zheng et al. (2025b) predicts whether a question is likely to yield a useful gradient and achieve a significant speedup with the prediction. Building on this second line of work, we extend dynamic sampling beyond efficiency. Rather than simply approximating full-dataset training by pruning low-gradient samples, our approach leverages response length as a heuristic to selectively retain data that contribute more to learning. This design improves not only training efficiency but also the final effectiveness of RLVR.

## 3 PRELIMINARIES

### 3.1 REINFORCEMENT LEARNING WITH VERIFIABLE REWARD

**Group Relative Policy Optimization** Group Relative Policy Optimization (GRPO) Shao et al. (2024) is a variant of Proximal Policy Optimization (PPO) Schulman et al. (2017) that removes the need for a separate critic model. In GRPO, the advantage is computed by sampling a group of responses for the same prompt, taking the average reward as the baseline, and then measuring each response's reward relative to this baseline. The objective function for GRPO is given as:

$$\mathcal{J}_{\text{GRPO}}(\theta) = \mathbb{E}_{(q,a)\sim\mathcal{D},\, \{o_i\}_{i=1}^{G}\sim\pi_{\theta_{\text{old}}}(\cdot|q)}$$

$$\left[ \frac{1}{G} \sum_{i=1}^{G} \frac{1}{|o_i|} \sum_{t=1}^{|o_i|} \Big( \min\big(r_{i,t}(\theta)\,\hat{A}_{i,t},\, \text{clip}(r_{i,t}(\theta), 1-\varepsilon, 1+\varepsilon)\,\hat{A}_{i,t}\big) - \beta\, D_{\text{KL}}(\pi_\theta \,\|\, \pi_{\text{ref}}) \Big) \right],$$

(1)

where,

$$r_{i,t}(\theta) = \frac{\pi_\theta(o_{i,t} \mid q, o_{i,<t})}{\pi_{\theta_{\text{old}}}(o_{i,t} \mid q, o_{i,<t})} \tag{2}$$

$$\hat{A}_{i,t} = \frac{r_i - \text{mean}\big(\{R_i\}_{i=1}^{G}\big)}{\text{std}\big(\{R_i\}_{i=1}^{G}\big)}. \tag{3}$$

**Decoupled Clip and Dynamic sAmpling Policy Optimization** Decoupled Clip and Dynamic Sampling Policy Optimization (DAPO) Yu et al. (2025) was proposed later and has become a solid baseline in the field due to its stability. Specifically, DAPO removes the KL-divergence term in the objective and introduces asymmetric clipping ranges, using different bounds for the lower and upper limits of the importance sampling factor. The optimization objective of DAPO is as follows:

$$\mathcal{J}_{\text{DAPO}}(\theta) = \mathbb{E}_{(q,a)\sim\mathcal{D},\, \{o_i\}_{i=1}^{G}\sim\pi_{\theta_{\text{old}}}(\cdot|q)}$$

$$\left[ \frac{1}{\sum_{i=1}^{G}|o_i|} \sum_{i=1}^{G} \sum_{t=1}^{|o_i|} \min\Big(r_{i,t}(\theta)\,\hat{A}_{i,t},\, \text{clip}\big(r_{i,t}(\theta),\, 1-\varepsilon_{\text{low}},\, 1+\varepsilon_{\text{high}}\big)\,\hat{A}_{i,t}\Big) \right] \tag{4}$$

$$\text{s.t.} \quad 0 < \big|\{\, o_i \mid \text{is\_equivalent}(a, o_i)\,\}\big| < G, \tag{5}$$

where $r_{i,t}(\theta)$ and $\hat{A}_{i,t}$ are the same as those used in GRPO, as shown in Eq. 2 and Eq. 3. In Eq. 5, DAPO applies a filter based on the current accuracy of the model, computing the loss only on prompts whose accuracy is neither 0 nor 1 in the current sampling batch. To maintain a fixed batch size, DAPO continues sampling until the batch is filled before performing training.

DAPO also introduces a buffer-length mechanism to penalize overlong responses, as shown below.

$$R_{\text{length}}(y) = \begin{cases} 0, & |y| \leq L_{max\_limit} - L_{\text{cache}} \\ \dfrac{(L_{max\_limit} - L_{\text{cache}}) - |y|}{L_{\text{cache}}}, & L_{max\_limit} - L_{\text{cache}} < |y| \leq L_{max\_limit} \\ -1, & L_{max\_limit} < |y| \end{cases} \quad (6)$$

Where $L_{max\_limit}$ denotes the maximum response length, and $L_{cache}$ is a tunable hyperparameter.

**Group Sequence Policy Optimization**  Besides DAPO, Group Sequence Policy Optimization (GSPO) is another well-acknowledged loss in the community. Specifically, GSPO introduces an alternative objective that applies clipping at the level of the entire response:

$$\mathcal{J}_{\text{GSPO}}(\theta) = \mathbb{E}_{x \sim \mathcal{D}, \{y_i\}_{i=1}^{G} \sim \pi_{\theta_{\text{old}}}(\cdot|x)} \left[ \frac{1}{G} \sum_{i=1}^{G} \min\Big( s_i(\theta)\, \hat{A}_i,\, \text{clip}(s_i(\theta), 1 - \epsilon, 1 + \epsilon)\, \hat{A}_i \Big) \right], \quad (7)$$

### 3.2 Role of Response Length in LLM Reasoning

Response length plays an increasingly important role in LLMs on reasoning-related tasks. On one hand, LLMs often rely on long reasoning chains, also known as Chain-of-Thought, to improve solution accuracy. As a result, responses in reasoning tasks are generally longer, creating a bottleneck for efficiency. This is especially evident in recent models, where self-reflection and extended reasoning are built-in capabilities that can be triggered autonomously; in such cases, response length serves as a proxy for the model's self-estimation of task difficulty.

Meanwhile, LLMs have been shown to suffer from an overthinking problem. In particular, Marjanović et al. (2025); Kumar et al. (2025) find that incorrect responses are, on average, longer than correct ones. Researchers have further proposed using response length as an important feature to predict whether a response is correct.

Taken together, response length in reasoning models serves both as an indicator of problem difficulty as perceived by the model and as a useful signal of response accuracy.

## 4 Length-aware Dynamic Sampling for Policy Optimization

In this section, we formally introduce **L**ength-aware dynamic **S**ampling for **P**olicy **O**ptimization (LSPO). We first describe the core filtering logic, and then explain how it is implemented in practice.

### 4.1 Length-aware Filtering

In prior work, dynamic sampling has largely focused on filtering out samples that do not provide gradients, or on predicting which samples are most likely to yield gradients in order to approximate full-data optimization. As a result, these approaches primarily improve training efficiency, while the performance of the converged models remains largely unchanged.

When examining sampled responses generated by LLMs, response length often reflects the model's perceived difficulty of a question. Short responses typically indicate high confidence, with little need for self-reflection. In contrast, longer responses suggest uncertainty, as the model frequently revisits or revises its reasoning path.

To develop efficient and effective reasoning models, we aim for LLMs to produce reasoning paths that are as short as possible while still yielding correct answers. Accordingly, in dynamic sampling, it is important to always include the shortest responses, since they represent the ideal case of confident and correct reasoning. At the other extreme, the longest responses are also valuable: they correspond to problems where the model invests significant computational effort. Training on these examples helps the model become more confident on difficult questions. Moreover, when responses are already among the longest, further training is more likely to shorten them rather than extend their length. Training on these samples can make the model more efficient. In contrast, responses of

intermediate length are less informative. They lack the certainty of short responses and do not consistently drive improvements in reasoning efficiency, making them less useful for targeted training.

To incorporate the above intuition into training, we formally propose Length-aware dynamic Sampling for Policy Optimization (LSPO). Specifically, let $L(q)$ denote the average response length by tokens:

$$L(q) := \frac{1}{G}\sum_{i=1}^{G}|o_i| \tag{8}$$

where $o_i$ is sampled from the distribution defined in Eq. 1 and Eq. 4. We further introduce the following auxiliary notations:

$$F_{L(q)}(t) := \Pr[\,L(q) \le t\,], \quad t \in \mathbb{R}, \qquad \text{(cumulative distribution function of } L(q)) \tag{9}$$

$$Q_{L(q)}(\alpha) := \inf\{\,t \in \mathbb{R} : F_{L(q)}(t) \ge \alpha\,\}, \qquad \alpha \in (0,1). \tag{10}$$

LSPO put an additional constraint on the questions considered:

$$L(q) \le Q_{L(q)}(L_{low}) \ \lor \ [L(q) \ge Q_{L(q)}(L_{high})]. \tag{11}$$

where $L_{low}$ and $L_{high}$ are two hyperparameters in the range of $(0,1)$ that control the filtering thresholds. A larger value of $L_{low}$ or a smaller value of $L_{high}$ filters fewer prompts and admits more data into training.

## 4.2 LENGTH-AWARE DYNAMIC SAMPLING

---

**Algorithm 1** Training Iteration in Length-aware dynamic Sampling for Policy Optimization (LSPO)

1: **Input:** Dataset $\mathcal{D}$; Default rollout batch size $B_r$; Training batch size $B_t$; Filtering threshold $L_{low}, L_{high}, L_{max}$; Number of rollout per prompt $G$;

2: $\mathcal{B} \leftarrow \varnothing$;                ▷ Rollout Stage
3: **repeat**
4:   Sample prompts $\{q_i\}_{i=1}^{B_r}$ from $\mathcal{D}$ until batch size $= B_r$;
5:   Rollout generation on $\{q_i\}_{i=1}^{B_r}$;
6:   Filter out zero-variance prompts in $\{(q_i,o_i)\}_{i=1}^{B_r \times G}$ as a new set $\{(q_i,o_i)\}_{i=1}^{B'_r}$;
7:   Calculate $L(q)$ from Eq. 8 on $\{(q_i,o_i)\}_{i=1}^{B'_r}$ ;
8:   Calculate $F_{L(q)}$ and $Q_{L(q)}$ from Eq. 9 and Eq. 10 on $\{(q_i,o_i)\}_{i=1}^{B'_r}$;
9:   Filter useful prompts based of Eq. 12 using $L_{low}, L_{high}, L_{max}$ as a final set $\{(q_i,o_i)\}_{i=1}^{B_r *}$;
                     ▷ Length-aware Filtering
10:   $\mathcal{B} \leftarrow \mathcal{B} \cup \{(q_i,o_i)\}_{i=1}^{B_r *}$;
11: **until** $|\mathcal{B}| \ge B_t$
12: $\mathcal{B} \leftarrow$ (Randomly) Select $B_t$ examples from $\mathcal{B}$;
13: Update actor model with RLVR algorithms, e.g. GRPO or DAPO, on $\mathcal{B}$;    ▷ RLVR Training

---

Empirically, we find that retaining prompts with the extreme longest responses can degrade performance. To address this, we introduce another hyperparameter $L_{max}$ to cap the maximum response length of the retained prompts, which modifies the constraint to:

$$L(q) \le Q_{L(q)}(L_{low}) \ \lor \ [L(q) \ge Q_{L(q)}(L_{high}) \land L(q) \le Q_{L(q)}(L_{max})]. \tag{12}$$

One challenge in filtering is that the distribution of response lengths is not known in advance. To address this, LSPO computes percentiles dynamically based on the samples in the current rollout batch. In each batch, LSPO recalculate the filtering threshold, and retains a fixed percentage of responses after accuracy-based filtering. While this increases the sampling time per step, it ensures that gradients focus on the most informative samples, making each training step more effective.

| Loss | Filter | AIME25 | Olympiad | Minerva | Avg. |
|------|--------|--------|----------|---------|------|
| \multicolumn{6}{c}{Qwen-2.5-Math-7B} ||||||
| GRPO | Acc-only | 21.5 | 48.5 | 42.5 | 37.5 |
|      | LSPO     | 23.2 | 49.6 | 43.3 | **38.7** |
| DAPO | Acc-only | 22.0 | 48.3 | 43.4 | 37.9 |
|      | LSPO     | 22.7 | 49.3 | 43.7 | **38.6** |
| GSPO | Acc-only | 21.9 | 50.0 | 43.5 | 38.5 |
|      | LSPO     | 22.5 | 51.0 | 44.0 | **39.2** |
| \multicolumn{6}{c}{Qwen3-4B-Base} ||||||
| GRPO | Acc-only | 20.0 | 49.6 | 44.9 | 38.2 |
|      | LSPO     | 22.0 | 49.3 | 46.7 | **39.3** |
| DAPO | Acc-only | 18.5 | 48.9 | 44.9 | 37.4 |
|      | LSPO     | 20.9 | 49.9 | 44.2 | **38.3** |
| GSPO | Acc-only | 18.0 | 48.0 | 45.7 | 37.2 |
|      | LSPO     | 22.0 | 51.1 | 45.8 | **39.6** |

Table 1: Performance (avg@32 in percentage) across three challenging math benchmarks. We train LSPO on two different models with different underlying base algorithms used as the loss function. Compared to the base algorithms, LSPO consistently delivers a stronger model.

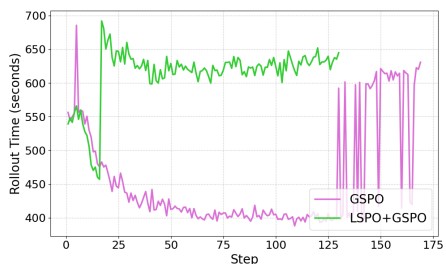

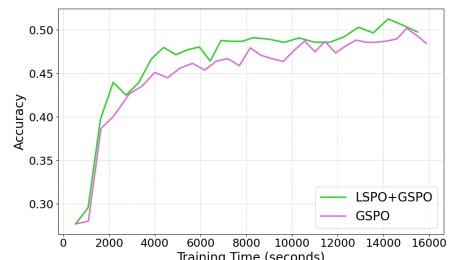

(a) The rollout time per-step.

(b) The avg@1 accuracy on Olympiad-bench regarding to the total training time used.

Figure 2: LSPO with GSPO as the base algorithms compared to GSPO itself trained on Qwen-2.5-Math-7B with DAPO-17K dataset and the training set of the MATH dataset.

Because the length threshold is recalculated in every rollout batch, our algorithm requires the batch size to be sufficiently large and the model to have a certain level of capability, so that enough prompts remain after the initial accuracy-based filtering. In practice, we find that the commonly used batch size of at least 256 works well for our algorithm.

We provide the pseudo-code of LSPO in Alg. 1. In each training iteration, we first discard prompts whose accuracy is uniformly 0 or 1, following prior work Yu et al. (2025), as they reduce sampling efficiency. We then compute the average response length for each remaining prompt and construct the corresponding distribution. Based on this distribution, we filter out prompts whose average response lengths fall in the middle, retaining only those in the shortest and longest ranges. The data pool for training is then updated with the prompts that pass this second round of filtering. This data collection process is repeated until sufficient samples are gathered, after which an arbitrary RLVR algorithm (e.g., GRPO or DAPO) is applied to the dataset $B$ to update the model. We refer to the RLVR algorithm that defines the loss as the base algorithm.

## 5 EXPERIMENTS

In this section, we empirically evaluate LSPO to demonstrate its efficiency and effectiveness as a meta–policy optimization algorithm for RLVR. The evaluation is organized as follows:

- In Sec. 5.2, we present the main results of LSPO across different models, datasets, and underlying training algorithms.

- In Sec. 5.3, we compare LSPO with alternative variants that dynamically retain samples based on length criteria, providing support for our current design choices.

### 5.1 EXPERIMENT SETUP

**Models and Datasets** In this paper, we evaluate our method using Qwen-2.5-Math-7B Yang et al. (2024) and Qwen3-4B-Base Yang et al. (2025). The models are trained on a combination of the DAPO-17K dataset Yu et al. (2025) and the MATH training set Hendrycks et al. (2021a). As noted in prior work Zheng et al. (2025b), training on DAPO-17K alone leads to performance degradation. To demonstrate that our approach is not limited to Qwen models, we additionally evaluate LSPO on Llama-3.2-4B-Instruct. Despite differences in architecture and training dynamics, LSPO continues to provide benefits on Llama. Due to space constraints, the detailed experiment settings and results for Llama are presented in the appendix.

For LSPO, we use a fixed hyperparameter of $L_{low} = 0.3, L_{high} = 0.65, L_{max} = 0.95$ by default.

**Evaluation** We train each model for 24 hours, saving a checkpoint every 10 steps, and report the average test accuracy from the last two checkpoints available at the time limit. To avoid potential data leakage Shao et al. (2025), we evaluate the trained models on three benchmarks: AIME-25 of America (2025), Olympiad He et al. (2024), and Minerva-Math Lewkowycz et al. (2022).

We report accuracy using 32 samples per prompt on the test benchmarks, i.e., avg@32. For training curves, we report accuracy based on a single sample per prompt, i.e., avg@1. All evaluation samples are generated with temperature of 1.0. Additional experimental details are provided in the appendix.

**Baselines and Base Algorithms** LSPO is a meta-heuristic for RL training, making it orthogonal to the underlying RL algorithms. In principle, LSPO can be combined with any RL algorithm to further improve training effectiveness. Accordingly, we use the base algorithms themselves as our only baselines, since other dynamic sampling methods such as GRESO do not improve effectiveness. In this paper, we focus on three state-of-the-art RL training algorithms as base algorithms: GRPO Shao et al. (2024), DAPO Yu et al. (2025), and GSPO Zheng et al. (2025a). By default, we adopt the overlong penalty and the asymmetric clipping scheme introduced in DAPO Yu et al. (2025). Hyperparameters are kept identical to those in the original baselines unless otherwise specified, with details provided in the appendix.

### 5.2 MAIN RESULTS

In Table 1, we present our numerical results. With LSPO, trained models perform better on nearly every benchmark tested, demonstrating the robustness of LSPO even without additional optimization of training speed. We leave further exploration of scalely improving training efficiency as future work, which we discuss later in the paper.

Figure 2 shows representative training curves comparing GSPO+LSPO with GSPO alone. Due to accuracy-based dynamic sampling, rollout time already varies even without LSPO. Under our current setting, where 60% of prompts are retained after accuracy filtering, the rollout cost increases by roughly 60%. However, we believe this overhead is generally worthwhile, since RL training often consumes about 80% of the total runtime to achieve the final 2% improvement. Thus, when accounting for total training time, LSPO not only provides a performance advantage but also improves efficiency in reaching the same level of model performance. The training curve in Fig. 2 clearly supports this conclusion.

| Filter Type | Kept Range | AIME25 | Olympiad | Minerva | Avg. |
|---|---|---|---|---|---|
| None | - | 22.0 | 48.3 | 43.4 | 37.9 |
| Accuracy Percentile | [0,30], [65, 95] | 22.2 | 48.4 | 42.7 | 37.8 |
| Length Percentile | [0, 60] | 22.4 | 47.8 | 42.8 | 37.7 |
| | [20, 80] | 22.0 | 47.5 | 42.5 | 37.3 |
| | [40, 100] | 21.5 | 48.4 | 43.0 | 37.6 |
| | [0, 30], [60, 90] | 22.5 | 47.9 | 43.0 | 37.8 |
| | [0, 30], [65, 95] | 22.7 | 49.3 | 43.7 | 38.6 |
| Length Value | [0, 30], [65, 95] | 21.4 | 47.2 | 41.8 | 36.8 |

Table 2: Ablation study with different filter designs trained on Qwen-2.5-Math-7B with DAPO. Avg@32 results are reported across benchmarks. Here we study filters applied in addition to the standard accuracy-based filter; thus, the "acc-only" filter in Table 1 corresponds to the *None* filter type shown here. The kept range is expressed in mathematical form to indicate the prompts retained after filtering. For example, "[0, 30], [65, 95]" with filter type "Length Percentile" means that we retain the shortest 30% of responses as well as the 65%–95% longest responses, which corresponds to $L_{low} = 0.3, L_{high} = 0.65, L_{max} = 0.95$ in Eq. 12.

## 5.3 Ablation Studies

In this section, we aim to address the following research questions regarding the design of LSPO:

- **RQ1.** Why does LSPO select the two extremes of average response length? Can training on the middle portion provide benefits?

- **RQ2.** LSPO is currently designed to retain a fixed percentage of data. Can a value-based threshold serve as an alternative?

- **RQ3.** Given the strong correlation between response length and accuracy in reasoning models, can filtering based on accuracy be effective?

To investigate these questions, we conduct ablation studies using Qwen-2.5-Math-7B. As before, all ablations are trained for 24 hours. We report the avg@32 accuracy results in Table 2.

**Training with other portions, especially the prompts with intermediate response length, is not effective.** Specifically, we fix the proportion of data retained in each sampling batch to 60%. During training, we observe that shorter responses generally receive higher critic scores, due to the correlation between response length and accuracy. However, models trained primarily on shorter responses fail to generalize these benefits to test sets. Conversely, when the shortest responses are excluded, performance slightly declines on both training and test sets.

These results suggest that training on both extremes of response length has complementary benefits. In contrast, training only on intermediate-length responses produces the worst performance and is strongly discouraged. Overall, while alternative parameter combinations of $L_{low}, L_{high}, L_{max}$ may yield further improvements, our current choice provides a strong and reliable baseline for LSPO. Since reasoning patterns are broadly similar across models, we believe the same parameter choices can be applied to a wide range of models and tasks without additional tuning.

**Training with value-based filtering is comparable to percentile-based filtering in performance but is less preferable due to its higher cost.** As an alternative to percentile-based filtering, one can use absolute values as the filtering criterion so that the number of responses kept in each batch is not fixed, unlike in DAPO and the current version of LSPO. However, we find that this design leads to substantially higher training costs. If the filtering value is an absolute length threshold, it is impractical to set such a threshold beforehand for a given task. Moreover, response lengths generated by the model vary significantly across different training phases. As a result, a fixed threshold often triggers repeated resampling in some steps while filtering almost no prompts in others.

An alternative is to use a relative value-based threshold. Specifically, we define the threshold for keeping prompts $T$ as a linear combination of the maximum and minimum lengths in the batch, $T = \alpha \min_q L(q) + (1 - \alpha) \max_q L(q)$, where $\alpha$ is a hyperparameter that controls the weighting. Similar to the percentile based threshold, we use three hyperparameters $\alpha_{low}, \alpha_{high}, \alpha_{max}$ to obtain three thresholds $T_{low}, T_{high}, T_{max}$. We keep prompts $q$ that satisfy the following constraint:

$$L(q) \leq T_{low} \vee [L(q) \geq T_{high} \wedge L(q) \leq T_{max}].$$

However, we empirically found that this approach suffers from the same issue: an indefinite number of resampling rounds during training. In particular, during the early and sometimes middle stages, nearly 20 resampling rounds were required, whereas percentile-based resampling typically required fewer than 10. As a result, relative value-based filtering leads to significantly longer sampling times, fewer training steps within the same budget, and ultimately worse evaluation performance compared to LSPO.

**Filtering based on accuracy is promising, but not as good as length for now.**   Similar to LSPO, samples can also be filtered based on accuracy obtained through verifiable rewards. This adds almost no additional cost, since an initial accuracy-based filter is already applied to remove prompts with zero variance. As shown in Table 2, while it does not greatly harm performance, it also provides no notable benefit compared to not filtering. Since filtering more samples increases the time required to collect rollouts during training, accuracy-based dynamic sampling is not preferable. However, if accuracy could be predicted in advance, for example by combining this variant of LSPO with methods such as GRESO, it may represent a promising direction for future exploration.

## 6 DISCUSSION

### 6.1 LIMITATIONS

LSPO focuses on average response length when filtering samples. Although it is designed as a meta reinforcement learning algorithm that can in principle be combined with any RLVR method, its applicability is not guaranteed for algorithms whose loss functions are strongly length dependent. Specifically, LSPO will always filter a portion of prompts, but whether this provides an advantage or at least preserves baseline performance is not theoretically guaranteed. In our experiments, however, we adopt the overlong buffer penalty introduced in DAPO Yu et al. (2025) across all loss functions, and LSPO empirically alleviates these concerns for now.

More broadly, LSPO is tailored to the dominant class of current RLVR algorithms; if post-training methods for LLMs evolve toward fundamentally different training dynamics, LSPO may no longer be applicable. For instance, if a model were trained to always generate a fixed number of tokens, such as exactly 4K per prompt, LSPO would be unable to perform meaningful filtering. Fortunately, most existing RLVR algorithms do not impose such constraints, and LSPO can therefore provide a benefit in practice for now on reasoning tasks.

### 6.2 FUTURE DIRECTIONS

**Improving Efficiency**   While LSPO currently does not substantially improve training efficiency, since it primarily filters prompts after responses are generated, it highlights a highly promising direction for extending dynamic sampling beyond simply removing samples that provide no gradients. Similar to how GRESO extends the dynamic sampling used in DAPO, future work could build predictors based on statistics from recent samples of the same prompt and filter out prompts whose predicted response lengths do not meet the criteria. We believe even a moderately accurate predictor could provide substantial speedups.

At the same time, while LSPO may not be directly compatible with RLVR algorithms that explicitly control response length in their loss functions connecting LSPO with models or algorithms that predict response length, such as LAPO Wu et al. (2025), offers an interesting direction. For example, during RL rollouts, the model could generate only the first sentence containing the predicted response length before selectively continuing only for prompts that satisfy the selection criteria. In this way, LSPO could gain the benefits of more informed filtering with even reduced rollout cost.

**Adaptive Filtering Threshold**   In this paper, we use a fixed percentile threshold to select the samples for training. While our ablation study examines the effect of different thresholds treated as hyperparameters and supports our current choice, we acknowledge that adaptively adjusting the threshold could surpass LSPO in both efficiency, measured by training speed, and effectiveness, measured by model capability. This represents a promising direction for future research.

**Alternative Filtering Criteria**   In this paper, we examined response length as the filtering criterion for dynamic sampling. Other signals, such as entropy or self-confidence, could serve as alternative criteria if properly designed for the task. We believe additional criteria may also improve performance, and whether they can match the gains achieved through length-based filtering remains an interesting question for future research.

## 7  CONCLUSION

In this paper, we introduced Length-aware dynamic Sampling for Policy Optimization (LSPO), a novel meta–reinforcement learning algorithm for LLM reasoning. LSPO computes the average response length during sampling and dynamically selects a subset of data for training. We evaluated LSPO across multiple models and datasets, demonstrating its effectiveness. Furthermore, as the first work to show that dynamic sampling can improve not only the efficiency but also the effectiveness of RLVR, we conducted a detailed ablation study on its design and outlined directions for future research. Overall, we hope our work serves as a stepping stone toward dynamic sampling strategies that move beyond gradient magnitude alone and incorporate a more diverse set of criteria.

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

## A  DISCLAIMER OF LLM USAGE

Apart from being the subject of study, large language models were used solely to polish the writing in this paper. All substantive contributions are attributable to the authors.

## B  EXPERIMENT DETAILS

**Hyperparameters**  In our paper, the experiments are mainly conducted using the same parameters as the recommended setting from DAPO Yu et al. (2025). Specifically, we use a mini batch size of 32, with training batch size of 512. We sample 8 responses for each question, and for the dynamic sampling, we sample $3\times$ the batch size for each round of dynamic filtering. We use a learning rate of $1e-6$ and a temperature of 1.0 during training. We do not use any top-p sampling, top-k sampling, or other sampling in either the training or evaluation loop. Considering Qwen-2.5-Math-7B and Llama-3.2-4B-Instruct only have 4096 context length, we set the maximum response length for experiments whose training set includes DAPO-17K dataset to be 2048, and 3072 for Llama-3.2-4B-Instruct on Math dataset, whose results will be provided later in the appendix. For all models, we have use the overlong penalty and the two clipping mechanisms introduced by

**Training**  Our code is implemented based on verl Sheng et al. (2024). All experiments are conducted on $4\times$Nvidia-H100 GPUs each. For prompts, we follow the GRESO Zheng et al. (2025b), to ask the models to think step by step and wrap the final answer in a box.

**Dataset**  In this paper, we have used DAPO-17K dataset and MATH dataset as the training set. DAPO-17K is introduced in Yu et al. (2025), which includes a total number of 17K datapoints from various sources of math problems. DAPO-17K has transformed all the answers into integers. MATH dataset Hendrycks et al. (2021b) is a slightly easier dataset, but the output format is not strictly as an integer, and include more diversity as latex form. We use math_verify to verify the results in the process. MATH dataset includes 7.5K training samples.

For evaluation, we have use AIME25 of America (2025), minerva-math Lewkowycz et al. (2022), and Olympiad-bench He et al. (2024). AIME25 is an Olympic-level dataset which was released in Fed 2025, and is known to be aside from the training set, given its late release date. AIME25 includes 30 problems in total, and all questions are asked in a special way so all answers are integers. Minerva-math includes 272 datapoints which includes Latex format. Olympiad-bench is introduced in 2024, and include multimodal questions. In this paper, we use the text part of the dataset, which includes 674 datapoints. The output format of Olympiad is even more diverse, and could include a list of numbers as results. Again, we use math_verify for checking the correctness of the answers.

**Prompt** Following previous work Zheng et al. (2025b), we use the prompt shown below, which requires the final output to be included in a box after thinking step-by-step.

> **Question Template**
>
> Please solve the following math problem: Question Description. The assistant first thinks about the reasoning process step by step and then provides the user with the answer. Return the final answer in \boxed{} tags, for example \boxed{1}. Let's solve this step by step.

## C ADDITIONAL EXPERIMENT RESULTS

### C.1 LLAMA-3.2-4B RESULTS

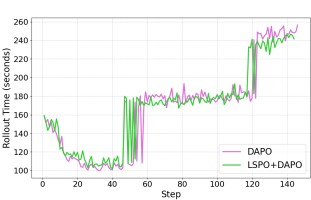

(a) The rollout time per-step.

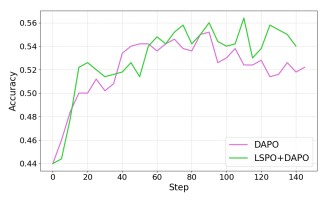

(b) The avg@1 accuracy on Math-500.

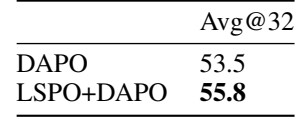

|  | Avg@32 |
|---|---|
| DAPO | 53.5 |
| LSPO+DAPO | **55.8** |

(c) Comparison of models trained for 24 hours using DAPO versus LSPO + DAPO tested on Math500.

Figure 3: LSPO with DAPO as the base algorithms compared to DAPO itself trained on the training set of MATH on Llama-3.2-4B-Instruct.

In addition to the Qwen series models used in the main paper, we have also train Llama-3.2-4B-Instruct. For Llama, because it is empirically performing worse and takes an extensively long time to finish the standard dynamic sampling process introduced by DAPO DeepSeek-AI et al. (2025) on DAPO-17K dataset, we train it with the training set of MATH dataset Hendrycks et al. (2021b) and evaluate it on Math-500, a subset of the test set of MATH introduced by OpenAI Lightman et al. (2023). We train the models for 20 epochs in total.

The training curve is shown in Fig. 3. We observe that LSPO consistently provides benefits as training approaches convergence, and the peak performance of models trained with LSPO surpasses that of models trained with DAPO. For Llama in particular, LSPO introduces almost no additional rollout cost, since most of the overhead is already absorbed by the accuracy-based filtering in DAPO.

### C.2 EXTENDING MAX RESPONSE LENGTH

Due to the high compute requirements and the limited availability of base models that support longer context windows, we followed prior work in the field Zheng et al. (2025b); Xu et al. (2025) in our

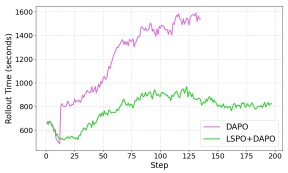

(a) The rollout time per-step.

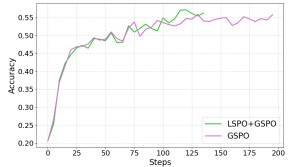

(b) The avg@1 accuracy on Olympiad-bench regarding to the number of training steps.

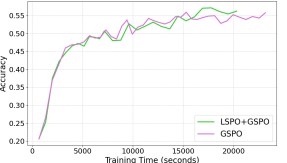

(c) The avg@1 accuracy on Olympiad-bench regarding to the total training time used.

Figure 4: LSPO with DAPO as the base algorithms compared to DAPO itself trained on Qwen3-4B with DAPO-17K dataset and the training set of the MATH dataset, and a max response length to 4096.

| Filter | AIME25 | Minerva | Olympiad | Avg. |
|---|---|---|---|---|
| Acc-only | 26.5 | 46.4 | 55.2 | 42.7 |
| LSPO | 28.2 | 46.6 | 57.0 | **44.0** |

Table 3: Avg@32 performance of Qwen3-4B-Base trained with the DAPO loss on 4K response length.

choice of context length in the main paper. All experiments were conducted under the same setting, ensuring that the comparisons remain fair.

As an illustration of the potential generalization of LSPO to longer context settings, we present results for training Qwen3-4B using the DAPO loss for 48 hours. As shown in Fig. 4 and Table 3, LSPO grows more slowly in the early stages but eventually surpasses the performance of standard training. This behavior arises because, when the model is far from saturation, the dominant factor is the number of training steps rather than the specific choice of data, which makes the early training trends of LSPO similar when measured in terms of training steps. However, as training progresses, prioritizing more informative and less noisy data begins to produce clearer benefits.

## C.3 ADDITIONAL TRAINING CURVES

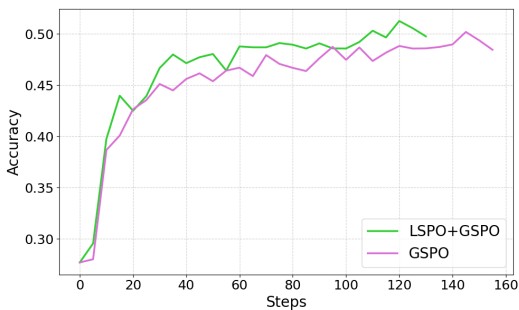

Figure 5: The avg@1 accuracy on Olympiad-bench regarding to the number of steps.

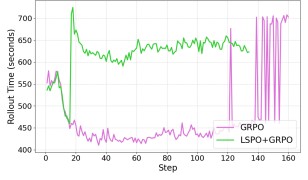
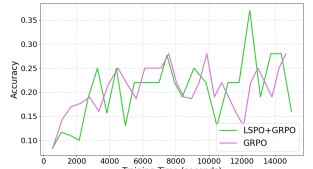
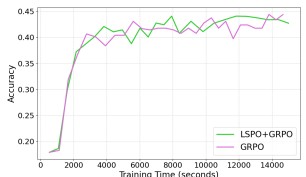

(a) The rollout time per-step.

(b) The avg@1 accuracy on AIME-25 regarding to the total training time used.

(c) The avg@1 accuracy on Minerva regarding to the total training time used.

Figure 6: LSPO with GRPO as the base algorithms compared to GRPO itself trained on Qwen-2.5-Math-7B with DAPO-17K dataset and the training set of the MATH dataset.

In the main paper, we provided training curves with the x-axis representing training time. In Fig. 5, we additionally show the training curves plotted against the number of steps. We observe that when the extra time used during rollout is not taken into account, LSPO exhibits a greater advantage compared to when training time is used as the x-axis.

However, we want to clarify that, in order to reduce the time spent on evaluation during training, all training curves are plotted using avg@1, which has substantially higher variance than the metrics reported in the tables. As a result, these curves serve primarily an illustrative purpose and are not a reliable basis for comparing the performance of large language models, especially in domains where the improvements are smaller. In Fig. 6, we additionally provide training curves using GRPO as the

underlying RLVR loss. We observe that LSPO combined with GRPO outperforms vanilla GRPO at most points during training.

