# OpenReview forum: "LSPO: Length-aware Dynamic Sampling for Policy Optimization in LLM Reasoning"
_ICLR.cc/2026/Conference — Submitted to ICLR 2026_

### Official Review · Reviewer_FWEL · 2025-10-29

**Soundness:** 2
**Presentation:** 3
**Contribution:** 2
**Rating:** 4
**Confidence:** 4

**Summary:**

The paper introduces LSPO, a reinforcement learning algorithm aimed at enhancing reasoning in LLMs. Motivated by the overthinking issue in LLMs, LSPO uses dynamic sampling based on response length to improve model effectiveness. It selectively retains prompts with the shortest and longest responses, focusing on data likely to be uncertain or overlength. The paper demonstrates that LSPO consistently improves the effectiveness of LLMs trained on reasoning tasks, including various challenging datasets and models. An ablation study is also provided to examine different length-based sampling strategies.

**Strengths:**

1. **Originality:** The perspective is fresh on dynamic data sampling in RLVR, using response length as a heuristic to guide training.
2. **Clarity:** The explanations of the algorithm and its components are clear, with well-documented experiment settings and discussions.
3. **Potential:** This work has the potential to influence future RLVR strategies and could lead to more efficient RL on LLMs.

**Weaknesses:**

1. **Training efficiency:** While LSPO improves the effectiveness of the model, its sampling process adds significant extra computation time per step, which could be a drawback in large-scale training scenarios.
2. **Response length limitation:** The filtering mechanism based solely on length may not generalize well if the model were required to generate fixed-length responses (e.g., for reasoning effort manipulation, creative writing, and other length-related tasks). Further work is needed to adapt LSPO to such constraints.
3. **Applicability to broader scopes:** The current formulation may not extend well to other types of RL tasks or environments where response length is not primarily correlated with performance. For instance, in coding tasks, writing Rust/C++ snippets inherently requires longer responses during reasoning compared to Python/JavaScript and other script-like languages.
4. **Limited interpretability:** The rationale behind why length-based sampling improves RL dynamics is not deeply explored. More analysis on the relationship between response length sampling and RL dynamics would strengthen the paper's contributions.

**Questions:**

1. What does $T_{min}$ stand for in Line 417?
2. How does RL dynamics change under the same training steps in Figure 2? I understand that controlling the training time is valid but not informative enough.

---

> ### Author Response · Authors · 2025-11-22
>
> Thank you for your detailed feedback. Here we address your concerns one by one:
> > Training efficiency
>
> While we acknowledge that our algorithm introduces additional computation time per step, our experiments already account for this, as all training runs are constrained to 24 hours of wall clock time. As you noted, our training curves are also plotted with respect to training time, ensuring a fair comparison.
> > Applicability to broader scopes
>
> We acknowledge that LSPO may not extend naturally to domains in which response length is not informative. However, we would like to point out that coding tasks do not necessarily pose a problem: Our thresholds are computed within each sample batch, and as long as the coding language within a batch is largely consistent, LSPO can still operate effectively.
> > What does $T_{min}$ stand for in Line 417?
>
> Thank you for pointing this out. This refers to the threshold used to retain the shortest responses; that is, a prompt $q$ is kept if $L(q) \le Tmin$. We have updated the draft to clarify this and have changed the notation from $T_{min}$​ to $T_{low}$​ for consistency with the $L_{\text{low}}$ used in the default percentile-based filtering.
> > How does RL dynamics change under the same training steps in Figure 2?
>
> As you noted in the first weakness you listed, our algorithm introduces additional computation time per training step. Therefore, when the x-axis is changed from training time to training steps, our algorithm shows an even larger improvement. We have updated the draft to include this result in the appendix.

---

### Official Review · Reviewer_CV7d · 2025-10-31

**Soundness:** 2
**Presentation:** 2
**Contribution:** 2
**Rating:** 2
**Confidence:** 4

**Summary:**

This paper proposes LSPO, a meta dynamic sampling algorithm for RLVR to improve reasoning performance of large language models. LSPO leverages response length as a heuristic signal and dynamically filters training examples, retaining prompts with either the shortest or longest sampled responses and discarding those in the middle.

**Strengths:**

- The algorithm is practical and compatible with any RLVR pipeline.
- Good discussion of limitations and future research.

**Weaknesses:**

- The approach is primarily heuristic, and the paper lacks a deeper theoretical explanation of why focusing on the shortest and longest responses drives stronger policy updates. More evidence or analysis is needed to justify this selection mechanism beyond empirical observation.
- The algorithm filters out prompts with medium length which could lead to under-training on those prompts.
- GSPO is used as a baseline throughout the paper, but the method is not introduced or sufficiently explained.
- For Llama-3.2-4B, only training curves are shown. No test-set evaluation results are provided, making the claim of cross-model generality incomplete.
- The experimental scale is relatively limited. Qwen models are capped at 2048 tokens and Llama at 3072 tokens, which is short for reasoning-focused RLVR. Since the core contribution relies on response length filtering, the short context limits the generalization of these findings to more realistic long-reasoning settings.
- Performance improvements are small (<2% in many cases). Additional trials with multiple seeds and reporting mean and std would help validate whether improvements are statistically significant rather than noise.
- Typos:
    - Equation 7 missing a summation sign.
    - Equation 10 missing ']'

**Questions:**

Please refer to the weaknesses.

---

> ### Author Response · Authors · 2025-11-22
>
> Thank you for your constructive feedback. Here we replies to your concerns one by one:
> >  More evidence or analysis is needed to justify this selection mechanism beyond empirical observation.
>
> Thank you for pointing this out. We acknowledge that our algorithm primarily provides a heuristic for training that is supported by empirical evidence, and that we do not offer strong theoretical guarantees. However, we would like to emphasize that the absence of a formal theoretical foundation is a common limitation at the current stage of LLM research. For example, prior works such as DAPO [1] and GRESO [4] also do not provide concrete theoretical justifications for their performance, yet they are well regarded within the community. For this reason, we focus on demonstrating the empirical effectiveness of our algorithm and leave a deeper theoretical investigation to future work.
> > The algorithm filters out prompts with medium length which could lead to under-training on those prompts.
>
> We would like to emphasize that training on all prompts does not guarantee better LLM performance. Several prior works have shown that selecting higher-quality or more informative data can lead to improved training outcomes [2, 5]. Our algorithm uses response length as a heuristic to dynamically select the prompts that are most beneficial for the current stage of training.
> > GSPO is used as a baseline throughout the paper, but the method is not introduced or sufficiently explained.
>
> Thank you for pointing this out. We have updated our paper draft to include the description of GSPO in the paper.
> > For Llama-3.2-4B, only training curves are shown.
>
> We would like to clarify that for Llama 3.2 4B Instruct, we trained on the MATH training set and evaluated on Math500. The training curves shown correspond to the avg@1 performance on Math500. As mentioned in the appendix, because the capabilities of the base models differ, we cannot evaluate them on the same test set used for the Qwen models. However, we agree that formally evaluating the trained model using avg@32, which has lower variance, strengthens the conclusions. We have added these results to Figure 3(c) in the updated version of the paper and have also included the comparison below. Once again, LSPO demonstrates clear benefits in terms of model capability.
> |           | Avg@32   |
> |-----------|----------|
> | DAPO      | 53.5     |
> | LSPO+DAPO | **55.8** |
>
> > The experimental scale is relatively limited.
>
> Due to the high compute requirements and the limited availability of base models that support longer context windows, we followed prior work in the field [3, 4] in our choice of context length in the original paper. All experiments were conducted under the same setting, ensuring that the comparisons remain fair. Because of time and resource constraints, we provide here an additional set of experiments on Qwen3 4B trained with DAPO using response lengths up to 4096 for 48 hours.
>
> We observe that LSPO grows more slowly at the beginning but eventually surpasses the performance of standard training. This occurs because, in the early stages when the model is far from saturation, the primary factor is the number of training steps rather than the specific choice of data. However, as training progresses, prioritizing more informative and less noisy data begins to yield clearer benefits.
>
> We have added a corresponding explanation and the associated results, both in tabular form and as training curves, to Section C in the appendix of the revised paper.
>
> > Additional trials with multiple seeds and reporting mean and std would help validate whether improvements are statistically significant rather than noise.
>
> Unfortunately, due to the substantial compute required for RLVR on large language models, we are unable to conduct multiple runs to report full statistical results. This limitation is common in the field, and most prior works likewise rely on a single seed [1, 3, 4]. In our paper, we report results not only for a single configuration but across multiple models and multiple underlying losses, demonstrating that our proposed method is stable. To further support reproducibility, we have included the code used in our experiments as supplementary material.
>
> > Typos
>
> Thank you for pointing them out. We have fixed them in the latest version of our paper.
> [1]. Yu Q, Zhang Z, Zhu R, et al. Dapo: An open-source llm reinforcement learning system at scale[C]. Neurips 2025.
>
> [2]. Zhou C, Liu P, Xu P, et al. Lima: Less is more for alignment[C].  Neurips 2023
>
> [3]. Xu Y E, Savani Y, Fang F, et al. Not all rollouts are useful: Down-sampling rollouts in llm reinforcement learning[J]. arXiv preprint arXiv:2504.13818, 2025.
>
> [4]. Zheng H, Zhou Y, Bartoldson B R, et al. Act Only When It Pays: Efficient Reinforcement Learning for LLM Reasoning via Selective Rollouts[C]. Neurips 2025.
>
> [5]. Li X, Zou H, Liu P. Limr: Less is more for rl scaling[J]. arXiv preprint arXiv:2502.11886, 2025.

---

### Official Review · Reviewer_VW63 · 2025-11-01

**Soundness:** 2
**Presentation:** 2
**Contribution:** 2
**Rating:** 2
**Confidence:** 4

**Summary:**

The paper proposes LSPO (Length-aware Sampling for Policy Optimization), a dynamic data sampling strategy for reinforcement learning with verifiable rewards (RLVR) in LLM reasoning. LSPO filters training prompts based on the average length of generated responses, retaining only those with the shortest or longest outputs under the hypothesis that these extremes are most informative for policy improvement. The method is evaluated across multiple base models (Qwen, Llama) and RLVR algorithms (GRPO, DAPO, GSPO), reporting modest but consistent gains in final accuracy on math reasoning benchmarks.

**Strengths:**

- The core idea—leveraging response length as a signal for dynamic data selection—is intuitive and grounded in prior observations about “overthinking” in LLMs.
- LSPO is designed as a meta-algorithm, making it compatible with various RLVR base methods, which enhances its practical applicability.
- The ablation studies systematically explore design choices (e.g., percentile vs. value thresholds, retained length ranges), providing useful empirical insights into the role of response length in RLVR training dynamics.

**Weaknesses:**

- Unclear problem formulation and unsubstantiated core assumption: The paper’s central motivation rests on the claim that “intermediate-length responses are less informative” and that filtering them improves final model effectiveness. However, this key assumption is never empirically validated. The ablation study shows that training only on intermediate-length responses yields lower performance, but this does not imply that including them in a full-data setting harms learning—yet the filtering strategy is justified precisely on that implicit premise. Without evidence that intermediate-length samples actively degrade training (e.g., via gradient conflict analysis, noise estimation, or ablation with full-data vs. filtered-data under matched compute), the rationale for discarding them remains speculative. Moreover, the paper does not clearly articulate what concrete learning problem LSPO solves: Is it combating overthinking? Improving sample efficiency? Enhancing generalization on hard examples? The lack of a well-defined objective makes it difficult to assess whether LSPO meaningfully advances the field or merely implements a heuristic with marginal gains.
- Insufficient comparison with prior dynamic sampling methods: The paper claims that existing dynamic sampling approaches (e.g., GRESO, PODS) only improve efficiency, not final model effectiveness, yet provides no experimental evidence to support this assertion. A direct comparison of LSPO against these methods in terms of final accuracy is missing.
- Marginal gains relative to overhead: The reported improvements are small, while LSPO incurs nontrivial rollout overhead. The cost-benefit trade-off is not convincingly justified.
- Incomplete empirical validation: Training curves are only shown for GSPO + LSPO vs. GSPO. Curves for GRPO and DAPO (the other two base algorithms) are absent, making it difficult to assess whether LSPO consistently accelerates learning or merely shifts the final performance slightly.
- Incomplete training dynamics analysis: The paper presents training curves for only two specific configurations: (1) Qwen-2.5-Math-7B with GSPO on Olympiad-bench, and (2) Llama-3.2-4B-Instruct with DAPO on MATH. However, the main results in Table 1 span two models, three base algorithms (GRPO, DAPO, GSPO), and three evaluation benchmarks (AIME25, Olympiad, Minerva). Without training curves for other combinations—e.g., GRPO+LSPO on AIME25 or DAPO+LSPO on Minerva—it is unclear whether LSPO consistently accelerates learning or merely yields small final gains under favorable conditions. Given the marginal absolute improvements (often ≤1%), this omission weakens the evidence for LSPO’s general effectiveness.
- Lack of theoretical justification: There is no analysis or proof suggesting that filtering by response length should improve convergence or generalization. The heuristic is plausible but not grounded in optimization or learning theory.
- Technical error in formulation: Equation (3) defining average response length $L(q) := \frac{1}{G} |o_i|$ lacks a summation which is ill-formed.

**Questions:**

- Problem clarity: The paper claims LSPO improves “final effectiveness” unlike prior dynamic sampling methods, but never specifies what concrete learning problem it solves—e.g., mitigating overthinking, improving hard-example generalization, or accelerating convergence. What is the precise objective that justifies filtering by length?
- Core assumption: The method assumes intermediate-length responses are “less informative,” yet the ablation only shows that training exclusively on them performs poorly. This does not prove they harm full-data training. Can the authors provide evidence (e.g., with matched compute) that excluding them actually helps?
- Gap Justification: Could the authors clarify what specific prior works in RLVR they consider as “dynamic sampling for efficiency only”? Are there no existing methods that use data selection to improve final accuracy? If such methods exist, how does LSPO differ in objective or mechanism?

- Theoretical Grounding: Is there any theoretical rationale (e.g., based on gradient variance, signal-to-noise ratio, or curriculum learning) that explains why filtering by response length extremes should improve policy optimization?

- Experimental Results:
  - Baseline comparison: The claim that methods like GRESO “do not improve effectiveness” is unsupported by experiments. Given LSPO’s marginal gains (often ≤1%), a direct comparison against GRESO/PODS in final accuracy is essential to validate its novelty.
  - The paper states that “other dynamic sampling methods such as GRESO do not improve effectiveness.” Is this claim supported by experiments? If not, could the authors include a direct comparison of LSPO against GRESO, PODS, or similar methods in terms of final test accuracy?
  - Training curve coverage: Training curves are shown only for two specific (model, algorithm, dataset) combinations. Can the authors provide curves for other main settings in Table 1 (e.g., GRPO+LSPO on AIME25) to support the claim of consistent improvement?

---

> ### Author Response · Authors · 2025-11-22
>
> Thank you for your time in our paper. Here we answer your questions one by one:
> > What is the precise objective that justifies filtering by length?
>
> Sampling at the prompt level is important because it allows the optimization process to focus on more informative data while reducing noise. This has been demonstrated in prior work on accuracy-based filtering and curriculum learning. In our paper, we incorporate a length-based filter because, as discussed in Section 3.2, response length is an informative statistic for reasoning tasks. Using it as a heuristic during training is therefore well motivated and reasonable.
> > Can the authors provide evidence (e.g., with matched compute) that excluding them actually helps?
>
> **In Table 1 of our paper, which presents our main results, we have compared LSPO with the accuracy-only filter.** The accuracy-only filter is the variant that does not exclude prompts with intermediate response lengths. All experiments are conducted under the same total training time of 24 hours, ensuring that the overall compute budget is matched across conditions.
> > Are there no existing methods that use data selection to improve final accuracy?
>
> To the best of our knowledge, no prior work has specifically focused on this aspect. We would be happy to offer further discussion on the differences between our method and any concrete reference you may provide.
> >  Is there any theoretical rationale (e.g., based on gradient variance, signal-to-noise ratio, or curriculum learning) that explains why filtering by response length extremes should improve policy optimization?
>
> Given that theoretical understanding of LLMs is still limited, we are unable to provide a formal theoretical analysis at this time. However, optimizing using extreme cases is generally known to benefit training by ensuring sufficiently large gradients and reducing noise, as discussed in prior work such as PODS [1].
> >  a direct comparison against GRESO/PODS in final accuracy is essential to validate its novelty.
>
> In our paper, we use exactly the same training data as GRESO. Below, we include the numbers reported in their work as a comparison to our proposed method. We observe that our method, LSPO, outperforms GRESO when both use the same GRPO loss.
> |       | Minerva  | Olympiad |
> |-------|----------|----------|
> | GRESO | 35.4     | 44.1     |
> | LSPO  | **43.3** | **49.6** |
>
> >  Can the authors provide curves for other main settings in Table 1 (e.g., GRPO+LSPO on AIME25) to support the claim of consistent improvement?
>
> We would like to clarify that the training curves are plotted using avg@1, which has substantially higher variance than the avg@32 metrics reported in the tables. As a result, these curves are intended primarily to illustrate training dynamics rather than to compare LSPO with other algorithms in terms of model capability. For this reason, we do not believe that adding additional training curves would be broadly informative. However, we have included the specific plot you requested in Section C of the appendix, which shows that LSPO can still provide a benefit most of the time.
>
> [1]. Xu Y E, Savani Y, Fang F, et al. Not all rollouts are useful: Down-sampling rollouts in llm reinforcement learning[J]. arXiv preprint arXiv:2504.13818, 2025.

---

### Official Review · Reviewer_1eiE · 2025-11-02

**Soundness:** 2
**Presentation:** 2
**Contribution:** 1
**Rating:** 2
**Confidence:** 5

**Summary:**

This paper introduces LSPO (Length-aware Sampling for Policy Optimisation) — a meta-algorithm for reinforcement learning with verifiable rewards (RLVR) in reasoning-focused LLMs. The method dynamically filters prompts during RL training based on response length, keeping the shortest and longest ones (based on percentile thresholds), motivated by empirical observations that overthinking correlates with longer, incorrect reasoning chains. LSPO aims to improve training effectiveness (final accuracy) rather than efficiency. Experiments and reported results show modest accuracy improvements over baselines.

1. The core idea of filtering samples based on response length is heuristic rather than theoretically grounded. Can the authors provide more intuition on why this is important?
2. Prior works such already explored length-aware optimisation, including direct regularisation of response length. LSPO’s novelty lies mainly in applying this as a data sampling heuristic, not as a new RL algorithm. The contribution feels incremental, more like a training trick than a principled new method.
3. Reported improvements are marginal. This makes me ask if such gains are within normal RL variance and could be due to stochasticity rather than the method itself. Can the authors conduct statistical significance tests to understand if those are actual gains?
4. Can the authors report results beyond math reasoning datasets? There are other reasoning domains; would we see such gains there as well?
5. I am not sure if 24 hrs is actually enough time for RL convergence. Can the authors please elaborate on the choice of this number? Why 24 hrs? Have you seen stable conclusions across the RL experiments?
6. From my understanding, the proposed method changes the batch composition per step? Is this true? If so, are the comparisons made under fixed time or batch settings? If so, it might be worth controlling those for the number of updates per unique sample.
7. To me, the claims are overstated. I wouldn't agree that this is the first paper to study the length-aware setup. I believe other algorithms have considered this already during optimisation.
8. The plots need to include variances. What stops me from saying those results in the figure are hand-picked? I think it is important to report variances.

**Strengths:**

Please see above

**Weaknesses:**

Please see above

**Questions:**

Please see above

---

> ### Author Response · Authors · 2025-11-22
>
> We thank the reviewer for the insightful, constructive feedback. Below, we address each of the points you raised.
>
> 1. > Can the authors provide more intuition on why this is important?
>
> Sampling at the prompt level is important because it allows the optimization process to focus on more informative data while reducing noise. This has been demonstrated in prior work on accuracy-based filtering and curriculum learning. In our paper, we choose to incorporate a length-based filter because, as discussed in Section 3.2, response length is an informative statistic in reasoning tasks. Using it as a heuristic during training is therefore well motivated and reasonable.
>
> 2. > The contribution feels incremental, more like a training trick than a principled new method.
>
> We respectfully disagree with the critique regarding novelty. Given the rapid development of RLVR algorithms, identifying techniques that further improve RLVR performance is highly valuable to the community. A prominent example is the success of DAPO, which is composed of several relatively small modifications, what could be described as training tricks, yet it has become a strong and widely adopted baseline. Similarly, **we argue that our paper provides a meaningful and solid step toward improving model performance**, and we believe the level of novelty is appropriate and sufficient.
>
> 3.  > Can the authors conduct statistical significance tests to understand if those are actual gains?
>
> > The plots need to include variances.
>
> Unfortunately, due to the substantial compute required for RLVR on large language models, we are unable to conduct multiple runs and report full statistical results. However, **this limitation is common in the field, and most prior works are also based on a single seed [1, 2, 3]. In our paper, we report results not only for a single configuration but across multiple models and multiple underlying losses.** To further support reproducibility, we have provided the code used in our experiments as supplementary material.
>
> 4. > Can the authors report results beyond math reasoning datasets?
>
> Unfortunately, due to the limited availability of sufficiently large training datasets in this area, and because applying our method to other domains such as coding would require additional resources to operate a sandbox environment for verifiable rewards, we are unable to provide such results within the rebuttal period. However, we would like to emphasize that this limitation is common in the field, and prior works have likewise reported results primarily on mathematical reasoning domains.
>
> 5. > Can the authors please elaborate on the choice of this number? Why 24 hrs? Have you seen stable conclusions across the RL experiments?
>
> We chose a 24-hour training window due to compute resource constraints. As shown in our training curves, the RL training largely saturates within this timescale. In addition, we extended training with DAPO on Qwen3-4B-Base and observed that the additional improvement was very limited. Moreover, all of our experimental settings follow the same time constraint, ensuring that the comparison remains fair.
>
> | Training Time | Filter   | AIME25 | Minerva | Olympiad-Bench | Avg. |
> |---------------|----------|--------|---------|----------------|------|
> | 24            | Acc-only | 18.5   | 48.9    | 44.9           | 37.4 |
> | 24            | LSPO     | 20.9   | 49.9    | 44.2           | 38.3 |
> | 48            | Acc-only | 19.0   | 49.2    | 44.6           | 37.6 |
> | 48            | LSPO     | 21.3   | 50.1    | 45.0           | 38.8 |
>
> 6. > From my understanding, the proposed method changes the batch composition per step? Is this true? If so, are the comparisons made under fixed time or batch settings? If so, it might be worth controlling those for the number of updates per unique sample.
>
> As you noted in point 5, all comparisons are conducted under a fixed budget of 24 hours of total wall clock time. Because we filter out samples whose current average response lengths are not at the two extremes, it is not feasible to control the number of updates per unique sample. This quantity naturally varies across prompts within a single LSPO run.
>
> 7. > I believe other algorithms have considered this already during optimisation.
>
> We respectfully request a clear reference for this point. Once provided, we would be happy to offer further discussion on the differences between our method and the papers you have in mind.
>
> [1]. Yu Q, Zhang Z, Zhu R, et al. Dapo: An open-source llm reinforcement learning system at scale[C]. Neurips 2025.
>
> [2]. Zheng H, Zhou Y, Bartoldson B R, et al. Act Only When It Pays: Efficient Reinforcement Learning for LLM Reasoning via Selective Rollouts[C]. Neurips 2025.
>
> [3]. Xu Y E, Savani Y, Fang F, et al. Not all rollouts are useful: Down-sampling rollouts in llm reinforcement learning[J]. arXiv preprint arXiv:2504.13818, 2025.

---

### Author Response · Authors · 2025-11-22
**General Response and Paper Draft Updated**

We thank all reviewers for the detailed and constructive feedback on our paper. In addition to providing responses to each individual question, we have updated the draft to include the following major revisions:
- An introduction to GSPO in the preliminaries.
- A clearer definition of the value-based ablation in the ablation study section.
- Avg@32 evaluation for testing LSPO on the Llama 3.2 4B Instruct model, added in Section C of the appendix.
- Additional training curves using GRPO as the underlying RLVR loss, included in Section C of the appendix.
- Additional results extending the maximum response length to 4096 tokens, also provided in Section C of the appendix.

All changes are highlighted in blue where possible. We invite you to review the updated version, and we are happy to address any further questions or concerns.

Thank you again for your time and effort in reviewing our work. We would greatly appreciate it if you could let us know whether our rebuttal has addressed your concerns and consider updating the scores accordingly.

---

### Meta-Review · Area_Chair_yHvs · 2026-01-06

**Summary:**

The decision to reject this paper is primarily driven by a strong consensus among Reviewers 1eiE, VW63, and CV7d regarding the lack of statistical robustness and the heuristic nature of the proposed LSPO method. The reviewers collectively argued that the reported empirical improvements are marginal and potentially insignificant. A critical concern shared by the committee is that without multi-seed variance analysis, these small gains cannot be reliably distinguished from standard training noise. Furthermore, Reviewer VW63 and others highlighted that the core contribution, using solution length as a reward signal, relies on an unvalidated heuristic. The paper fails to provide a rigorous theoretical justification or a clean ablation to explain why penalizing mid-length trajectories is causally effective, distinct from simple batch or training duration effects. Consequently, the reviewers viewed the work as an incremental engineering tweak rather than a substantive methodological advance suitable for publication.

**Reviewer Concerns:**

Regarding the rebuttal, the authors made commendable efforts to address concerns related to completeness and reproducibility. They successfully expanded the experimental scope to include the Qwen model family and longer context lengths up to 4096 tokens, effectively satisfying the specific requests from Reviewers CV7d and VW63 regarding the limited initial scope. Additionally, the clarification regarding the fixed 24-hour wall-clock training time effectively resolved Reviewer FWEL's concerns about computational overhead and efficiency. However, the most critical concerns regarding validity and soundness remain outstanding. Despite adding breadth to the experiments, the authors did not provide the multi-seed variance analysis explicitly required by Reviewer 1eiE to prove the robustness of the marginal gains. Similarly, the rebuttal failed to isolate the causal mechanism of the "mid-length" hypothesis through the requested theoretical grounding or matched-compute proofs, leaving the fundamental question of the method's validity unanswered.

**Reviewer Scores:**

If the reviewers had participated fully in the post-rebuttal discussion, I project that the majority of scores would remain below the acceptance threshold. Reviewer 1eiE's score would likely remain a Reject (2) because their specific requirement for variance analysis to validate the marginal gains was not met, leaving the core barrier to acceptance intact. Similarly, Reviewer VW63's score would likely remain unchanged (2) as the fundamental objection regarding the lack of theoretical justification for the heuristic persists despite the added baselines. Reviewer CV7d might raise their score slightly to a 4, acknowledging the extensively improved experimental scope (adding Llama-3 and Qwen results), but the remaining lack of theoretical robustness likely prevents a shift to acceptance. Conversely, Reviewer FWEL would likely move to a weak Accept (6) since their primary reservations about efficiency overhead were clearly resolved; however, this positive shift represents a minority view based on operational factors and does not outweigh the fundamental validity concerns held by the rest of the committee.

---

### Decision · Program_Chairs · 2026-01-26

Reject